# Nano-Perforated Silicon Membrane with Monolithically Integrated Buried Cavity

**DOI:** 10.3390/mi16010104

**Published:** 2025-01-16

**Authors:** Sanjeev Vishal Kota, Anil Thilsted, Daniel Trimarco, Jesper Yue Pan, Ole Hansen, Jörg Hübner, Rafael Taboryski, Henri Jansen

**Affiliations:** 1DTU Nanolab, National Centre for Nano Fabrication and Characterization, Technical University of Denmark, Ørsteds Plads B347, 2800 Kongens Lyngby, Denmark; 2Spectro Inlets Aps, Generatorvej 6A, 2860 Søborg, Denmark

**Keywords:** monolithic, buried cavity, perforated membrane, silicon nanofabrication, nano-pores, DREM, high aspect ratio, CORE

## Abstract

A wafer-scale process for fabricating monolithically suspended nano-perforated membranes (NPMs) with integrated support structures into silicon is developed. Existing fabrication methods are suitable for many desired geometries, but face challenges related to mechanical robustness and fabrication complexity. We demonstrate a process that utilizes the cyclic deposit, remove, etch, and multi-step (DREM) process for directional etching of high-aspect-ratio (HAR) 300 nm in diameter nano-pores of 700 nm pitch. Subsequently, a buried cavity beneath the nano-pores is formed by switching to an isotropic etch, which effectively yields a thick NPM. Due to this architecture’s flexibility and process robustness, structural parameters such as membrane thickness, diameter, integrated support structures, and cavity height can be adjusted, allowing a wide range of NPM geometries. This work presents NPMs with final thicknesses of 4.5 µm, 6.5 µm, and 12 µm. Detailed steps of this new approach are discussed, including the etching of a through-silicon-via to establish the connection of the NPM to the macro-world. Our approach to fabricating NPMs within single-crystal silicon overcomes some of the limitations of previous methods. Owing to its monolithic design, this NPM architecture permits further enhancements through material deposition, pore size reduction, and surface functionalization, broadening its application potential for corrosive environments, purification and separation processes, and numerous other advanced applications.

## 1. Introduction

Nano-perforated membranes (NPMs) are characterized by their nano-scale pores and perforated structure, which are critical for a broad spectrum of technological applications. Their high surface-area-to-volume ratio, tunable pore sizes and shapes, mechanical strength, chemical resistance, selectivity, and hydraulic permeability allow for versatile applications in industrial processes. In water and wastewater treatment, they are used to reduce contaminants like bacteria and heavy metals [1]. The food and beverage industry utilizes NPMs to purify products by eliminating microorganisms, thereby extending shelf life without undermining quality [2]. In the chemical and petrochemical sectors, NPMs facilitate efficient separation of gas molecular mixtures, boosting process efficiency [3].

Additionally, in biological applications like tissue engineering and regenerative medicine, large-scale NPMs support the cultivation of more cells simultaneously to achieve therapeutic outcomes [4] and enable the precise modulation of drug diffusion rates [5]. Finally, NPMs may also be used for high-pressure electroosmotic pumping [6]. These multifaceted properties make NPMs essential for advancing technology and improving industrial processes.

Several methods to fabricate NPMs using thin-film deposition and etch techniques with different materials have been developed [7,8,9]. Tong et al. [10], created a 10 nm thin SiN_x_ membrane with two-level support, consisting of a wet-etched wafer-thick Si frame carrying a 1000 nm thick micro-sieve with 5 μm diameter holes. Pores as small as 25 nm were etched into the 10 nm membrane using focused ion beam (FIB) etching, as illustrated in Figure 1A. Alternatively, Unnikrishnan et al. [11] presented a wafer-scale thin-film transfer process via fusion bonding and time-controlled plasma etching to fabricate a free-hanging perforated oxide nano-membrane on top of a supporting micromachined Si micro-sieve as illustrated in Figure 1B.

Another approach—less cumbersome than the above techniques—involved SOI (silicon-on-insulator) wafers [12], as demonstrated by Sainiemi [13], in Figure 1C. In this method, nano-pores are patterned and etched in a thin Si device top layer, and the thicker Si handle wafer is etched from the backside using the buried oxide layer (BOX) as an etch stop. The membrane is then released by HF wet etching of the SiO_2_. While this approach allows tight control of membrane thickness and the buried oxide layer, it restricts design flexibility and potentially delaminates the oxide layer under (thermal) stress. Thus, the non-monolithic architecture of the above NPM designs makes them susceptible to structural failure, particularly for large-area thin membranes that are required to achieve low-flow resistance [14]. This rules out the application of such membranes in environments where mechanical robustness is essential, and influences the choice of membrane material. Tackling a few of these shortcomings, Sato et al. [15] fabricated a silicon-on-nothing (SON) membrane structure based on the microstructure transformation of Si, as shown in Figure 1D. The process involved etching trenches and annealing them in a hydrogen atmosphere at high temperatures, promoting Si migration and forming an empty-space-in-silicon (ESS), i.e., a buried cavity. The free-hanging top layer formed is the SON layer in which pores can be etched to form the NPM, although this is not discussed in the paper. However, the formation and thickness of the ESS remain constrained by trench dimensions, thereby lacking geometric flexibility. For clarity, these fabrication methods for NPMs are shown in Figure 1.

While the fabrication approaches mentioned above have proven effective in creating NPMs, they also reveal limitations in fabrication robustness, geometric flexibility, and mechanical stability. One promising method that increases geometric flexibility and NPM strength is called buried channel technology (BCT), demonstrated by de Boer et al. [16]. This process utilizes the etching of trenches into Si using Si_x_N_y_+Cr as a hard mask, followed by sidewall protection with SiO_2_ or Si_x_N_y_ while keeping the mask intact. The protective layer at the trench bottoms is then directionally etched, forming a buried channel underneath trenches by wet or dry isotropic Si etching. However, this method cannot be directly applied in our cleanroom due to the potential contamination of oxide or nitride furnaces by wafers previously processed with metal or metal oxides. As a result, the hard mask materials must be removed after feature (trench or pore) etching (explained later in more detail). Due to these constraints, an alternative approach is necessary to effectively protect the top surfaces and edges while selectively removing the bottom protection layer to form buried channels or cavities. By introducing modifications to pore arrangements and adapting fabrication steps, our method enables the formation of buried cavities with integrated support structures using nano-pores. Additionally, the proposed approach offers flexibility for adaptation in other facilities with similar contamination sensitivity.

This study introduces and describes a process for the monolithic fabrication of a robust suspended NPM with integrated support structures in Si. It combines the DREM (deposit, remove, etch, and multi-step) or CORE (Clear, Oxidize, Remove, and Etch) processes (previously demonstrated for HAR etching) [17,18] for nano-pore etching with the BCT approach to form a buried cavity. Additionally, it includes etching a centrally aligned through-silicon-via (TSV) on the backside of the wafer to funnel flux from the NPM through a buried cavity for downstream processing. Consequently, this method overcomes structural and geometrical constraints, producing large-area NPMs with a wide range of thicknesses, which is an advantage for tailoring membranes to application requirements. Additionally, NPMs with support structures in single-crystal silicon ensure uniform stress distribution, maintaining mechanical strength and effectively addressing the challenges typically associated with NPM fabrication.

**Figure 1 micromachines-16-00104-f001:**
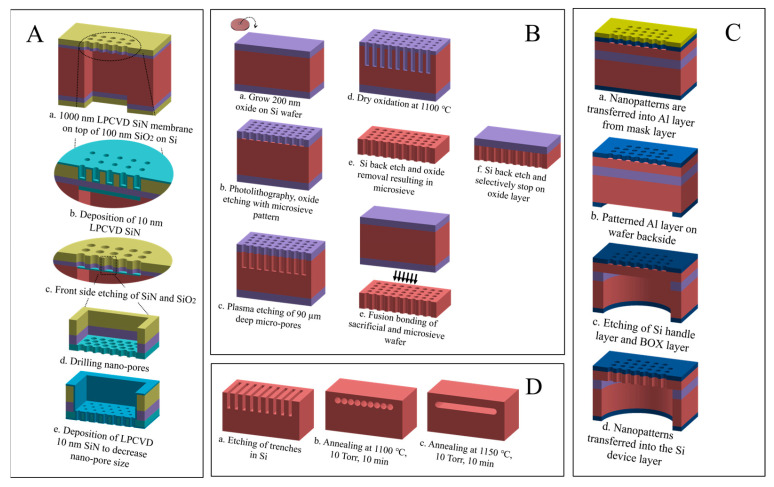
Illustrates process flows of some typical previous techniques to construct the NPM: (**A**) nanosieve supported by a micro-sieve supported by a <110> Si frame [10], (**B**) thin-film transfer of a wafer-scale nanosieve on a wafer-scale micro-sieve [11], (**C**) silicon-on-insulator approach [13], and (**D**) silicon-on-nothing approach [15].

## 2. Materials and Methods

A number of 150 mm Si wafers (Czochralski, double-sided polished, 500 ± 20 μm thick, <100>, 1–20 ohm-cm, boron-doped *p*-type, Siegert Wafer GmbH, Aachen, Germany) are used in this study. The fabrication process consists of a series of steps as shown in Figure 2:

(a) Alumina deposition on Si: Alumina is deposited using the atomic layer deposition (ALD) process (Picosun R200 ALD system, Picosun Oy, Espoo, Finland) to form a highly etch-resistant hard mask for the subsequent HAR Si etch in step d. Trimethyl aluminum (TMA) and water are used as precursors to create a chemical reaction on the wafer surface and deposit alumina in cyclic mode (TMA, purge, water, purge, TMA…). The deposition is performed for 500 cycles at 200 °C to deposit an alumina layer of 50 nm. The wafer is not pre-treated before depositing the alumina.

(b) Patterning nano-pores by deep UV (DUV) lithography: The wafer is spin-coated (Süss Gamma 2M spin-coater, Süss MicroTec, München, Germany) with a 65 nm bottom anti-reflective layer (BARC, DUV42s-6, Brewer Science Ltd., Paris, France) followed by 360 nm DUV resist (KRF M230Y, JSR-Micro, Leuven, Belgium). The wafer is then patterned using a DUV stepper (FPA-3000EX4, Canon, Tokyo, Japan) equipped with a projection lens (factor 5 reduction) with a numerical aperture of 0.6 and a 248 nm KrF excimer laser. A hexagonal array of circular nano-pores, each with a diameter of 300 nm and a donepitch of 700 nm, is patterned onto the resist using an exposure dose of 370 J/m^2^ with zero focus offset. Following exposure, the resist is developed in AZ 726 MIF (MicroChemicals GmbH, Ulm, Germany) for 60 s. This process results in a 7 mm wide perforated circular area featuring patterned nano-pores. Pore-free regions, each 10 µm in diameter, are intentionally designed and created in this circular perforated area, and they are repeated in hexagonal arrangement with 50 µm pitch to serve as support structures for the NPM.

(c) DUV pattern transfer into alumina: The process continues with transferring the DUV resist pattern into the BARC and alumina layers using an inductively coupled plasma (ICP) etch tool (ICP Metal Etch, SPTS Technologies Ltd., Newport, UK). For the 65 nm BARC, 8 min plasma etch is performed using a 10 sccm O_2_ gas flow at 1 mTorr, 0 °C, and 20 W platen power, resulting in 35 V DC self-bias. The pattern transfer into the alumina layer uses a 30 sccm BCl_3_ gas flow at 1 mTorr, −20 °C, 30 W platen power, and 450 W coil power for 240 s. The low platen power is specifically optimized to minimize resist erosion (only 360 nm thick) and achieve a straight wall profile in the alumina layer. The etching is followed by stripping the remaining photoresist and residues for 45 min in a plasma asher using 400 sccm O_2_ and 70 sccm N_2_ at 1000 W (300 Plasma processor, PVA TePla America Inc., Corona, CA, USA).

(d) HAR etching of nano-pores in Si: The Si HAR etching is performed using a deep reactive ion etching (DRIE) tool (Pegasus, SPTS Technologies Ltd., Newport, UK). A modified DREM process [17,18] is used to etch HAR pores with a straight profile using the alumina mask pattern. This is followed by another O_2_ plasma ashing process as previously described to remove residues caused by the Si etch. The remaining alumina is removed by etching in 10% hydrofluoric acid (HF) for 60 s followed by RCA cleaning.

(e) Protection of nano-pores by thermal oxidation: The 25 nm SiO_2_ sidewall protection of the nano-pores is achieved through dry oxidation of Si in a high-temperature tube furnace (Tempress) at 1050 °C for 10 min in O_2_.

(f) Capping of the surface by PECVD oxide: A non-conformal PECVD (Plasma-Enhanced Chemical Vapor Deposition) capping oxide layer is applied to reinforce the top surface, including the pore edges (explained later in more detail). The 340 nm PECVD oxide is deposited at 300 °C with a gas composition of 12 sccm SiH_4_, 1420 sccm N_2_O, and 392 sccm N_2_, at 700 mTorr and 150 W RF power (Multiplex PECVD system, SPTS Technologies Ltd., Newport, UK).

(g) Removal of oxide from nano-pore bottom: A directional etch is employed using SF_6_ plasma at 1.5 mTorr. The low-pressure environment is needed to generate active ions with high energy and a large mean free path, thus enabling the directional bombardment of the bottom of the nano-pores without harming the sidewall.

(h) Forming of the NPM: The formation of the NPM uses a dry etch approach, where F-radicals that reach the bottom of the pore undercut Si isotropically, eventually connecting adjacent pores to form a buried cavity. For this etching, a gas flow of 100 sccm SF_6_ is used at 1000 W coil power and 25 mTorr at 0 °C and without any platen power.

(i) Removal of oxide layers: The remaining thermal and PECVD oxide layers are removed by etching in 10% HF for 120 s.

(j) Deposition of etch-stop layer: A 50 nm layer of ALD Al_2_O_3_ is deposited to act as an etch-stop layer, preventing damage to the nano-pore membrane during the via etch in step j.

(k) Etching of TSV: The TSV of 50 µm in diameter is patterned on the wafer’s backside on a spin-coated (Gamma 2M, Süss MicroTec, München, Germany) 4 µm thick layer of AZMiR 701 (MicroChemicals, Ulm, Germany) positive resist using a maskless aligner (MLA) with 405 nm UV light (MLA150, Heidelberg Instruments GmbH, Heidelberg, Germany). The exposure parameters are set to 500 mJ/cm^2^ dose and zero defocus, the resist is post-baked at 110 °C for 60 s, and then a puddle develops for 120 s using AZ 726MIF, resulting in centrally aligned TSV to each circular nano-pore membrane. Subsequently, the developed pattern is transferred into the alumina layer using chlorine plasma as previously described. Finally, the TSV is etched down to the alumina etch-stop layer using the DREM process in a Si DRIE tool (Pegasus, SPTS Technologies Ltd., Newport, UK).

(l) Removal of etch-stop layer: As a final step in the fabrication process, the alumina etch-stop layer is removed by immersing the wafer in 10% HF (Aq) for 60 s, resulting in finished NPM with TSV in single-crystal Si.

For characterization, the wafer is manually cleaved into pieces. The cross-sections of the samples are then examined by scanning electron microscopy (SEM, Supra VP40, Zeiss, Jena, Germany) to analyze and verify the structural outcomes of the etching processes in detail. The hexagonal arrangement of the pores improves the probability of cleaving through some of them, thereby facilitating visual inspection of the structures.

## 3. Results and Discussion

This section presents the NPM and TSV formation results while providing detailed explanations of the critical process steps involved in forming the buried cavity.

### 3.1. HAR Etching of Nano-Pores in Si (Step D)

By optimizing the DREM process, the recipe shown in Table 1 results in HAR nano-pores, as shown in Figure 3. After 123 DREM cycles, resulting in *d_max_* = 13.0 ± 0.1 µm deep pores, the original 300 ± 10 nm pore openings exhibit a widening midway to become ca. *W_mid_* = 440 nm. This diameter is considered as the pore diameter for calculating the aspect ratio: *AR* = *d_max_*/*W_mid_* = 30. The scallop’s height is ca. *d_max_*/cycles ≈ 100 nm per cycle along the pores.

### 3.2. Etching of the Buried Cavity for Forming NPM (Step E–H)

An essential step in the buried cavity formation is to protect the Si sidewalls during the isotropic undercutting. The protection layer must be mechanically stable, be pinhole-free, and have a much lower etch rate than Si. Various layers such as LPCVD (low-pressure chemical vapor deposition) silicon nitride, ALD alumina, plasma-deposited fluorocarbon (FC) polymer, or plasma and thermal Si oxidation might be considered. In particular, the top convex corner of the pores needs proper protection as these locations erode much faster in plasma environments than in other places [16]. Several techniques might be helpful to obtain better top protection, and three are mentioned in [16], as shown in Figure 4a–c. However, these proposals all rely on an LPCVD SiN or other mask still present on top of the pores. In this study, the top mask (alumina) is removed in HF before the thermal oxidation of silicon, so a different strategy is needed. Hence, we initially opted for FC deposition because this technique intrinsically protects the top side due to the non-conformal nature of the FC plasma deposition (Figure 4d). Furthermore, the etching of nano-pores, its sidewall passivation, bottom removal, and an isotropic etch can be completed in the same plasma reactor within 20 min, allowing the entire sequence to be performed in one go (multi-step single-run) [19].

FC protection: A plasma FC layer is initially used to protect the sidewalls of 5 µm deep (*AR* = 16) nano-pores as it is non-conformal and easy to clear at the bottom of the pore [19,20]. The FC layer is deposited in 2 min using a flow of 100 sccm C_4_F_8_ at 12 m Torr and 1000 W coil power. Next, a low-pressure RIE process with a 40 sccm SF6 flow at 65 W platen power is applied for 1 min to remove the bottom FC through physical ion bombardment. Finally, a 7 min isotropic plasma etch with a flow of 100 sccm SF_6_ at 25 mTorr and 500 W coil power is introduced, forming the buried cavity without any sidewall damage (Figure 5a). However, FC passivation proved to be unreliable and inconsistent in providing coverage and protection, primarily due to conductance and reactant transport limitations that restricted the distribution of CFx radicals within the nano-pore interior [19]. Consequently, although a buried cavity was formed, sidewall damage to the pores was observed, as shown in Figure 5b. For HAR nano-pores or taller buried cavities, the required longer etch duration further compromises the structural integrity, leading to membrane collapse. In addition, the repeatability of FC deposition depends on the reactor’s condition (and history) and previous processes that significantly affect the outcome. Understanding and controlling these variables is crucial to improve the reliability of the process. As a result, while FC polymer can be effective for protecting sidewalls in lower-AR structures, it may not provide sufficient stability and reliability for HAR nano-pores on a wafer scale. Therefore, more robust sidewall protection is needed, which is covered in the next section.

SiO_2_ protection: To create SiO_2_ protection in nano-pores, many techniques are available that either form conformal (e.g., thermal, chemical, plasma, or LPCVD oxide) or non-conformal (e.g., spin-on glass or PECVD oxide) films. As learned in the previous section, non-conformal protection is likely to cause rupture of the sidewall down the pore. However, sole uniform protection cannot be used either as the top side etches faster than the bottom side during the bottom removal. Therefore, this paper proposes the best of both worlds: starting first with a dense oxide that uniformly protects the pore walls and subsequent addition of a non-conformal layer to give the top side extra protection to withstand the longer etch time while clearing the pore bottom (Figure 4e).

For the first uniform protection, we have a few options. Forming the oxide layer in the same plasma reactor would be ideal, but the self-limited oxide thickness (around 2–3 nm can be grown by room temperature plasma oxidation) is insufficient [21]. Rapid thermal oxidation (RTO) is also considered but is limited to 20 nm oxide layers due to chamber protection time constraints at high temperatures. This thickness is insufficient for the required versatile NPM fabrication platform. Therefore, thermal Si oxidation is chosen for sidewall protection due to its ability to form a controllable uniform SiO_2_ layer. Figure 6a shows 25 nm of thermal oxide protecting the nano-pore.

The double protection continues with PECVD oxide deposition, which is inherently non-conformal in nano-pores due to ion-enhanced reactions at high pressure (700 m Torr), resulting in frequent collisions that reduce the reactive species mobility and hinder the diffusion of reactive species into narrow spaces. As a result, more oxide is deposited near the pore opening, while the sidewalls and bottom of the pore have much less oxide, as shown in Figure 4e. Therefore, PECVD oxide of 340 nm is selected, as shown in Figure 6b (visualized with a yellow outline), on top of the 25 nm thermal oxide.

Continuing the NPM fabrication process, with double protection, the bottom oxide layer is removed using a low-pressure SF_6_ plasma (explained later in more detail), leaving behind a thin PECVD oxide layer as shown in Figure 6c, and an isotropic etch is performed for 17 min using 100 sccm SF_6_ flow at 25 m Torr and 1000 W coil power, to form a buried cavity that is 5 µm tall and correctly forming the NPM (Figure 6d). The Si etch rate at the bottom of the pore depends on the flux of etchant species that reach the bottom, which depends on the opening width and depth of the nano-pore: the well-studied RIE lag [22]. The time needed to undercut the NPM successfully is relatively short for low-AR nano-pores but increases with increasing pore AR. Nevertheless, it is possible to achieve NPM with different cavity depths, as illustrated Figure 6e,f.

In addition, a two-level nano-pore membrane with varying dimensions can be formed by repeating the presented process, as depicted in Figure 7. In the first phase, 5 µm deep nano-pores are etched, and a 1.5 µm tall, buried cavity is formed (Figure 7a–d). In the second phase, double oxide protection is applied, and a thermal oxide layer is grown to protect the nano-pore sidewalls and the cavity, while a PECVD oxide is used as a surface capping layer (Figure 7e). Following this, RIE with SF_6_ plasma is used to create openings at the bottom of the buried cavity through the narrowed nano-pores, and then 3.5 µm deep nano-pores are etched (Figure 7f). Subsequently, another double oxide protection step is applied to protect the sidewalls of the nano-pores and cavity (Figure 7g), followed by the selective bottom removal of oxide and formation of a second buried 1 µm tall cavity (Figure 7h); the resulting two-level NPM is shown in Figure 6g. This demonstrates the robustness and versatility of the process, providing a foundation for creating even more complex multi-level NPMs.

Before all the thermal oxidation steps, the wafer underwent a plasma ashing and RCA process. The etching times for nano-pores and cavity formation were finely tuned to achieve respective depths, while the thermal and PECVD oxide layers were kept constant at 25 nm and 340 nm, respectively.

### 3.3. Etching of TSV

TSV facilitates the transport of all flux collected in the buried cavity from the NPM to the macro-world specifically for downstream processing, allowing for seamless integration with larger-scale processes. The etching of a 50 µm in diameter TSV, which corresponds to an AR of 10, is straightforward, and the process parameters are listed in Table 2, which gives an etch rate of nearly 3.5 µm per cycle. The alumina layer in the buried cavity acts as an etch-stop layer and protects the suspended NPM, as seen in Figure 2k.

### 3.4. Fabrication Challenges

Several key challenges were encountered throughout the fabrication process that affected the robustness of the NPM fabrication, particularly with HAR nano-pores. These challenges are summarized below.

PECVD oxide deposition and bottom oxide removal. One major challenge involved managing the PECVD oxide thickness. Although it is possible to deposit thicker oxide layers, clogging of the nano-pores begins when the oxide thickness exceeds 360 nm, thus preventing the bottom clearance. Additionally, the non-conformal deposition (the so-called blooming effect) narrows the nano-pore opening, allowing only a limited flux of ions to pass through, as depicted in Figure 8a. However, the advantage of this blooming effect is that the PECVD oxide also shields the thermal oxide at the very top from off-normal ions from the imperfect ion angular distribution or deflections by the image force [22]. Then, as the etching progresses, the top oxide is gradually etched away, increasing the nano-pore opening and allowing ions to reach the sidewalls, as illustrated in Figure 8b. So, the bottom removal step is not uniform, as illustrated in Figure 8c. But, despite this incomplete removal, the opening created in the bottom protection is sufficient for F-radicals to etch the bulk Si and form the buried cavity. Any remaining bottom protection is subsequently removed during the cavity formation, ensuring effective continued etching.

Thermal oxide thickness. Another challenge was selecting an appropriate thermal oxide layer thickness. We found that less than 20 nm of the first uniform thermal oxide is insufficient during the buried cavity formation, which results in pinholes at the top and eventually leads to an NPM fracture. Conversely, clearing layers of more than 30 nm of thermal oxide needs more etch time than the PECVD can withstand and would require repetitive cycles of step f followed by step g in Figure 1. Thus, selecting the appropriate oxide combination (25 nm oxide and 340 nm PECVD oxide) is crucial to ensure the integrity of the NPM. But even then, the bottom oxide removal step should be tuned correctly, as explained next.

Ion-enhanced etching for bottom oxide removal. Clearing the bottom oxide posed another set of challenges. Initially, the experiments were performed at 0 °C, for 380 s, with a 40 sccm SF_6_ gas flow at 1.5 mTorr and 120 W platen power. However, the high power led to removing the top oxide and silicon etch at the surface before the bottom oxide was cleared, as shown in Figure 9a. Therefore, the power was reduced to 100 W while maintaining the other parameters. The bottom oxide was removed while leaving 10–15 nm PECVD oxide, but this led to silicon sidewall etching during step h. Increasing the etching time with 100 W platen power would clear the bottom, but the 10–15 nm oxide layer is too thin and could be etched away in under 30 s, risking membrane damage.

The platen power was further reduced to 85 W and etched for the same duration. This prevented damage to the top, while the bottom oxide layer was thinned but not entirely removed, leaving enough PECVD oxide to protect the membrane during the extended etching process. The etching time was subsequently adjusted to 465 s to achieve optimal results for a nano-pore with a depth of 13 µm to etch 25 nm bottom thermal oxide. This left 85–90 nm PECVD oxide with the capping oxide etch rate of ca. 30–35 nm/min. This procedure avoided unwanted etching at the top edges of the pores, as shown in Figure 6c. The difference in the etch rate of surface and bottom oxide is due to RIE lag in the HAR structure [22]. Additionally, reduced nano-pore opening caused by the PECVD oxide layer (Figure 8a) resulted in a lower initial etch rate for the bottom oxide.

However, when the platen power was further reduced to 75 W, the bottom oxide was only etched in some pores and remained unetched in others after 465 s (Figure 9b), indicating that a longer etch time would be needed. Although extending etch time could potentially resolve this issue, this was not tested due to concerns about thinning the capping oxide at the top edges of the nano-pores, which could compromise the structural integrity of the nano-pores. Therefore, precise tuning is essential for fabricating HAR NPMs. Our observations also revealed that as the membrane depth decreased from 13 µm to 9.5 µm and then to 7 µm, the required bottom oxide removal times reduced significantly, from 465 s to 275 s and finally to 180 s. This suggests that the etching rate approximately triples as the depth is halved, highlighting the need to carefully adjust etching parameters to ensure structural integrity and process consistency for the HAR NPM fabrication process.

While these conclusions hold for directional nano-pores, removing the bottom oxide layer from positive tapered nano-pores with a depth of 13 µm posed additional challenges, particularly when the bottom nano-pore diameter approached or fell below 50 nm. For instance, after growing a 25 nm thermal oxide, the bottom diameter was further reduced, effectively blocking incoming ions, reducing etch rate, and preventing oxide layer removal during the 465 s removal time, which resulted in no cavity formation.

NPM integrity, as revealed in the SEM analysis of Figure 6d (yellow dotted line), showed that the bottom part of the opened nano-pores underwent upward undercutting due to the isotropic etching nature of F-radicals [20]. This effect increased with increasing buried cavity height and effectively shrank the thickness of the NPM slightly, resulting in a final thickness of 12 µm; i.e., 1 µm of nano-pores experienced upward undercutting while forming a 5 µm tall, buried cavity. In addition to this, the choice of a 700 nm pitch for the 300 nm nano-pores was intentional. With a larger pitch between nano-pores, the isotropic etching beneath each nano-pore would take longer before a completely buried cavity formed. This extended etching time would likely necessitate enhanced sidewall protection to prevent erosion, making careful management of the protection layers critical for maintaining the structural integrity of the NPM.

On the other hand, smaller pitches reduce the distances between pores, and the inherent widening in the middle of the pore and under-etching could affect the structural integrity of the NPM for HAR nano-pores. Therefore, a 700 nm pitch was deemed reasonably optimal for 300 nm pores. Ultimately, the final NPM thickness was determined by the isotropic etch time, the buried cavity’s height, and the nano-pore pitch.

### 3.5. Mechanical Strength of NPM

The mechanical strength of the NPM is a critical characteristic and can be evaluated using various methods. For fluidic applications such as separation and filtration, the mechanical strength can be effectively represented by the transmembrane pressure (Δp). Using the maximum transmembrane pressure before membrane fracture with a radius, $r_{NPM}$, and membrane thickness, $t_{NPM}$, the fracture can be estimated using the following equation [10].(1)$$\Delta p_{max}=0.29K\frac{t_{NPM}}{r_{NPM}}\sigma_{yield}\sqrt{\frac{\sigma_{yield}}{E}}\;,$$
where $\sigma_{yield}$ is the yield stress and E is Young’s modulus of Si. K is the non-perforated fraction of the NPM (~0.5). However, while Young’s modulus of Si is well known (130 GPa), the yield stress may vary by orders of magnitude, and in addition depends on processing conditions. Hence, by choosing a very conservative (low) estimate for the yield strength of 165 MPa, a lower bound for $\Delta p_{max}\sim$2 bar is obtained, where a radius of 45 µm is also inserted for $r_{NPM}$ instead of 3.5 mm (actual membrane radius). A radius of 45 µm is the estimated radius of the largest unsupported region, where the support structures are removed for TSV etching. In this unsupported region, the membrane is more prone to fracture.

## 4. Conclusions

Nano-perforated membranes (NPMs) with monolithically integrated support structures were reliably fabricated in Si by leveraging the DREM process in combination with BCT. This study thoroughly investigated and emphasized the critical importance of each step by choosing a combination of thermal and PECVD oxide for surface protection to achieve pinhole-free sidewalls in nano-pores and ensure defect-free surfaces while forming the NPMs. Using an etch-stop layer during the through-silicon-via (TSV) etching process provided control, ensuring uniformity and consistency in membrane thickness and structural integrity. These improvements with standard fabrication technologies enhanced the overall fabrication process, paving the way for the consistent mass production of TSV-integrated NPMs on a wafer scale.

This monolithic fabrication approach effectively addressed key challenges commonly associated with existing modular methods, including complex alignment, fixed membrane thickness, choice of material, and mechanical stability. Furthermore, this method offers some geometric flexibility and mechanical strength with integrated support structures, allowing the production of NPMs with varying thicknesses and large areas. The TSV-integrated NPM demonstrated it is particularly well suited for applications requiring steady flux delivery through a single well-defined path. For instance, as an interface for mass spectrometry, the NPM serves as a capillary stop type of liquid/gas interface facilitating the separation of volatile gas molecules dissolved in the bulk liquid, which can then be analyzed using mass spectrometry.

Moreover, the process supports the creation of diverse patterns on the backside of the wafer, enhancing the functional capabilities of the membranes. To further broaden the applications of NPMs, they can be encapsulated using technologies such as ALD, thermal processes, and molecular vapor deposition to tailor electrical and wetting properties. Overall, this work presents a promising pathway for NPM fabrication, offering versatility and potential for various applications.

## 5. Outlook

As explained in the Section 3, rapid thermal oxidation (RTO) was initially explored as a promising alternative for protecting nano-pore sidewalls during isotropic silicon etching. The process showed some initial success, demonstrating potential in forming an oxide layer for sidewall protection. However, due to time constraints associated with the current RTO process at high temperature, which achieves only a maximum oxide layer thickness of 20 nm, it is insufficient for adequate sidewall protection during prolonged isotropic etching processes. Therefore, further development of this approach was not pursued. Nevertheless, the initial results from this process are presented here to highlight its potential as a sidewall protection method. The process outlined in Figure 10 is like the earlier method but uses the CORE process instead of DREM for etching nano-pores and RTO for sidewall protection instead of double oxide layers.

The process starts with transferring the DUV pattern into a 100 nm alumina layer using chlorine plasma, as shown in Figure 11a. Nano-pores are then etched to a depth of 3.75 µm using the CORE process [12] at room temperature, where oxygen acts as the self-limiting inhibitor, as shown in Figure 11c. Since the CORE process is FC-free, the sample can proceed directly to the rapid thermal processing tool (Annealsys AS-Premium V2) without requiring special cleaning treatments. A 40 min RTO at 1200 °C, 12 mbar, and 1500 sccm O_2_ results in a uniform 20 nm oxide layer, as depicted in Figure 11d. The alumina mask is retained during RTO, which serves as a capping layer and removes the need for non-conformal oxide capping. Figure 11a,b show the nano-pore-patterned alumina before and after RTO. After RTO, the amorphous alumina is transformed into crystalline alumina, and a transition usually happens above 800–900 °C. This change could increase the porosity of the alumina mask layer due to nanocrystal formation. However, the effect is more significant in thin films (2–7 nm) [23] than in thicker layers like the 100 nm alumina used here. Moreover, this change did not have an observed effect on the etching of Si in our experiments.

This is followed by removing the bottom oxide utilizing low-pressure SF_6_ plasma etching without damaging the nano-pores. Afterwards, a second DRIE of Si (Figure 10g) is performed using the CORE process with the alumina mask to etch the silicon to an additional depth of 500 nm. Finally, an undercut is performed using 1200 sccm SF_6_ at 220 mTorr with 2000 W coil power for 5 min, forming the cavity as shown in Figure 11e. DRIE etching of nano-pores is consistent across the wafer, ensuring uniform depth. Implementing a second DRIE step to etch an additional 500 nm into the Si provides better control over the cavity height. This additional anisotropic etch reduces the required duration of isotropic etching as it only needs to undercut Si between the pores; however, some upward undercutting is still observed. Despite this, the process improves sidewall integrity and facilitates more precise and uniform cavity formation.

With further optimization of RTO capabilities, while maintaining uniform oxide coverage, this approach offers a more sustainable and reproducible method for fabricating NPMs. Additionally, this method simplifies fabrication, reduces tool usage, and minimizes cross-contamination. It also avoids process drifting in the etching tools commonly associated with conventional methods using C_4_F_8_ for passivation, ensuring greater consistency and reliability in the final product. However, as stated before, the current RTO restrictions do not allow us to grow SiO_2_ layers of more than 20 nm. We need at least 25 nm for the intended mass spectroscopy application.

## Figures and Tables

**Figure 2 micromachines-16-00104-f002:**
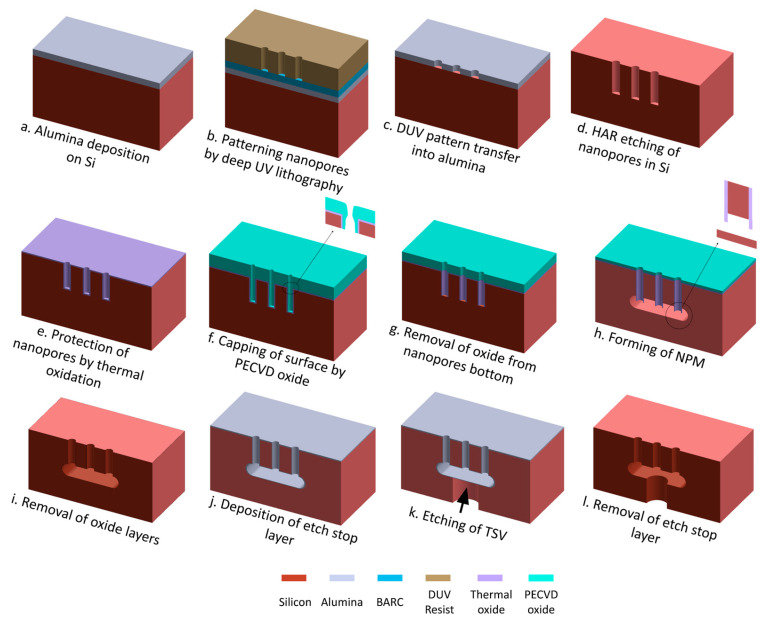
Demonstration of nano-perforated membrane formation in single-crystal Si. (**a**–**l**) A 3D schematic illustration showcasing the cross-sectional view of the fabrication process flow for forming a nano-perforated membrane with a through-silicon-via. Only three nano-pores are shown for clarity.

**Figure 3 micromachines-16-00104-f003:**
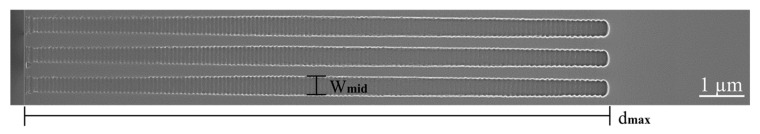
SEM image showing the high-aspect-ratio nano-pores with a directional profile. *W_mid_* = 440 nm is the maximum diameter at ca. half the depth of the nano-pore, and *d_max_* = 13 µm is the maximum depth of nano-pores, giving *AR* = *d_max_*/*W_mid_* = 30.

**Figure 4 micromachines-16-00104-f004:**
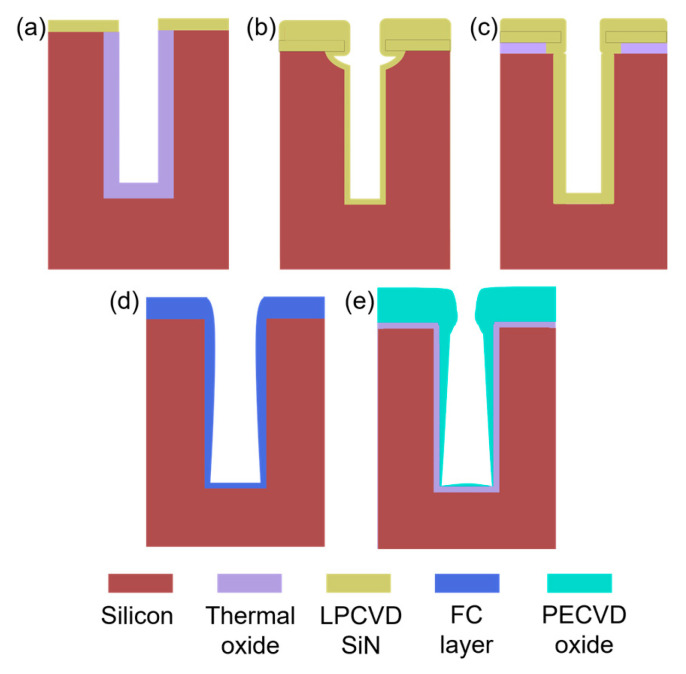
Three methods, (**a**–**c**), from [16] to protect the coating in the trench and two alternative ways, (**d**,**e**), used in this study for nano-pores. (**a**) Under-etching plus thermal oxidation, (**b**) isotropic pre-etching, (**c**) short sacrificial layer etching, (**d**) non-conformal fluorocarbon deposition in DRIE etch tool for nano-pores, (**e**) thermal oxidation topped with non-conformal PECVD oxide in nano-pores.

**Figure 5 micromachines-16-00104-f005:**
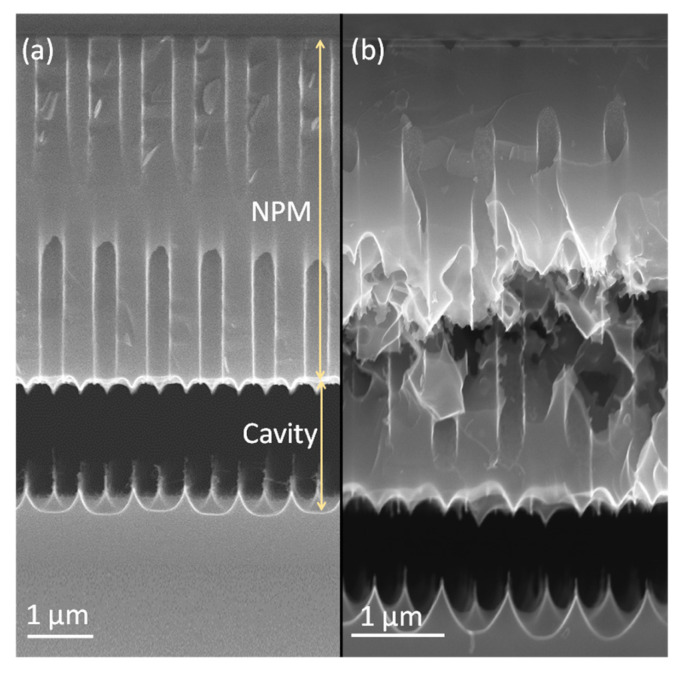
Impact of FC protection on nano-pore sidewall integrity with an increasing aspect ratio. Cross-sectional view of (**a**) successful suspended nano-perforated membrane formation with effective FC polymer protection; (**b**) inadequate FC protection at increased nano-pore depth, resulting in significant sidewall damage during suspended nano-perforated membrane formation.

**Figure 6 micromachines-16-00104-f006:**
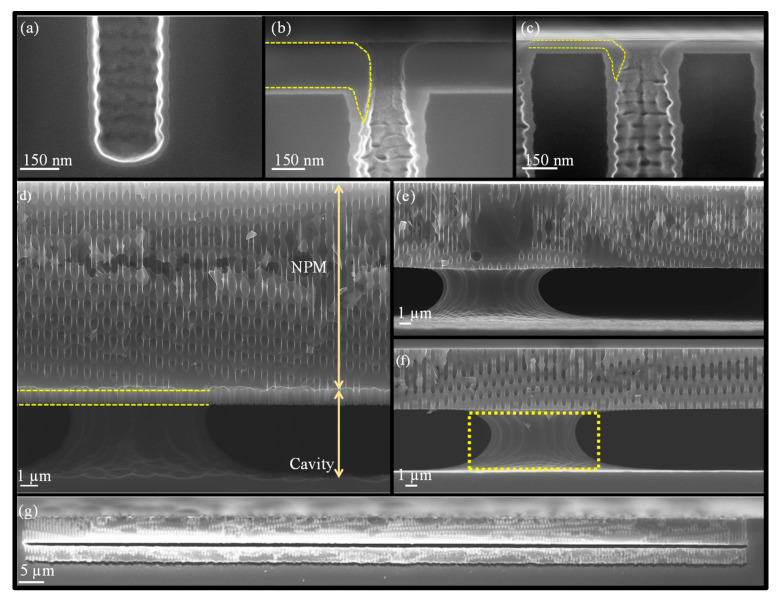
A cross-sectional overview of SEM images at distinct steps of buried cavity formation. (**a**) A close-up at the bottom of the Si nano-pore with a 25 nm conformal thermal oxide layer; (**b**) a close-up of the non-conformal plasma oxide step-coverage near the nano-pore opening highlighted by a yellow dotted line; (**c**) the top part of the nano-pore with remaining oxide after bottom clearance; (**d**) the 5 µm deep buried cavity formed underneath a 12 µm deep nano-perforated membrane; a clean cleave is difficult due to presence of the buried cavity, which is prone to irregular fracturing. This can also be seen in e and f: (**e**) a 6.5 µm nano-pore membrane with a 5 µm buried cavity; (**f**) a 4.5 µm nano-pore membrane with a 5 µm buried cavity, and the support structure highlighted by the black dotted line; and (**g**) a two-level nano-pore membrane which was formed by repeating the fabrication sequence twice.

**Figure 7 micromachines-16-00104-f007:**
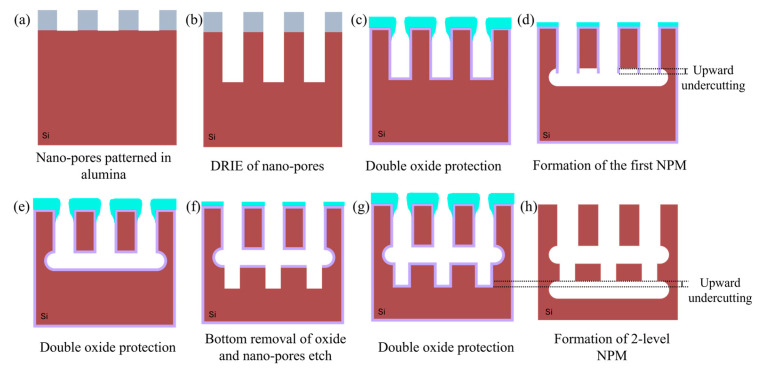
Schematic illustration showcasing the cross-sectional view of the fabrication process flow for forming a two-level nano-perforated membrane.

**Figure 8 micromachines-16-00104-f008:**
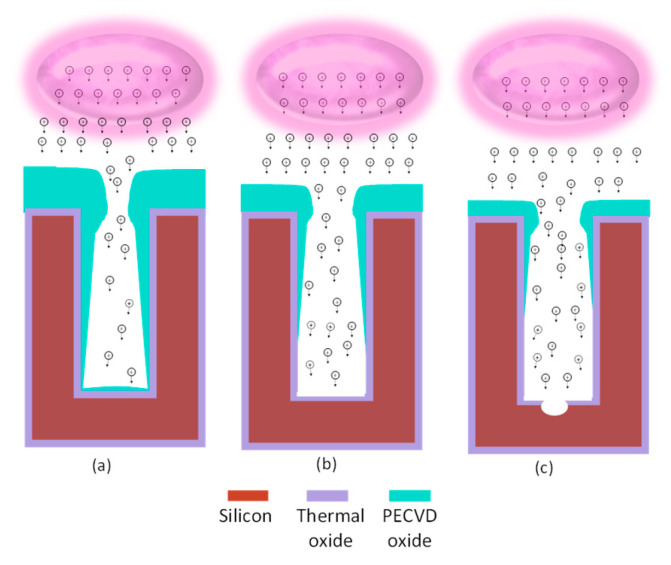
Illustrating the significance of reduced pore opening by non-conformal PECVD oxide in controlling the transport of high-energy ions into the pore to etch the bottom thermal oxide layer. (**a**) Reduced pore diameter directs ions through the middle of the pore, protecting the top edge and sidewalls; (**b**) increased opening width allows more ions to reach the bottom, with some lower-energy ions moving close to sidewalls that do not have sufficient energy to cause significant damage; and (**c**) further increased width improves ion spread, allowing more ions to reach the bottom with high directionality. In contrast, some ions are lost to the sidewalls and cause damage.

**Figure 9 micromachines-16-00104-f009:**
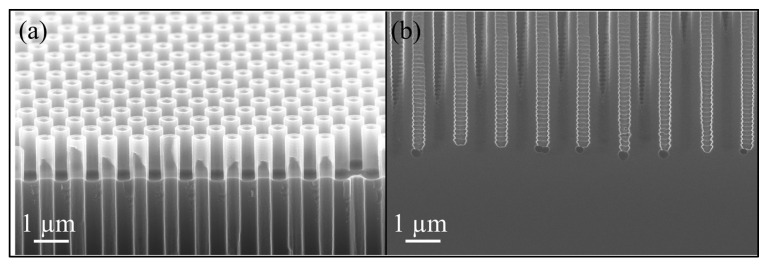
SEM images show the effects of platen power on oxide and silicon etching. (**a**) Etching at a high platen power of 120 W led to complete oxide removal at the surface and unintended silicon etching underneath, in a 20° tilt view; (**b**) etching at a low power of 75 W resulted in non-uniform removal of a bottom oxide layer.

**Figure 10 micromachines-16-00104-f010:**
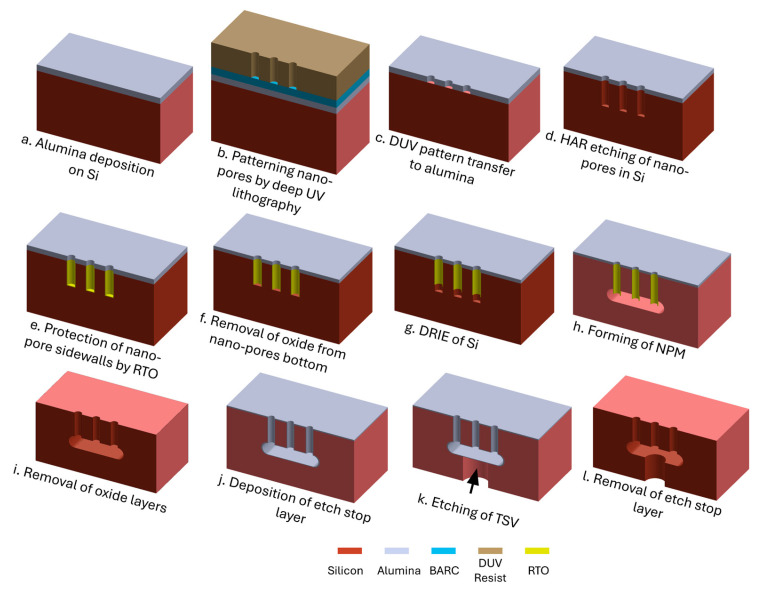
Demonstration of nano-perforated membrane formation in single-crystal Si with a through-silicon-via using the CORE process and rapid thermal oxidation. (**a**–**l**) A 3D schematic illustration displaying cross-sectional views of the Si at various stages of the fabrication process. For clarity, only three nano-pores are depicted for illustration.

**Figure 11 micromachines-16-00104-f011:**
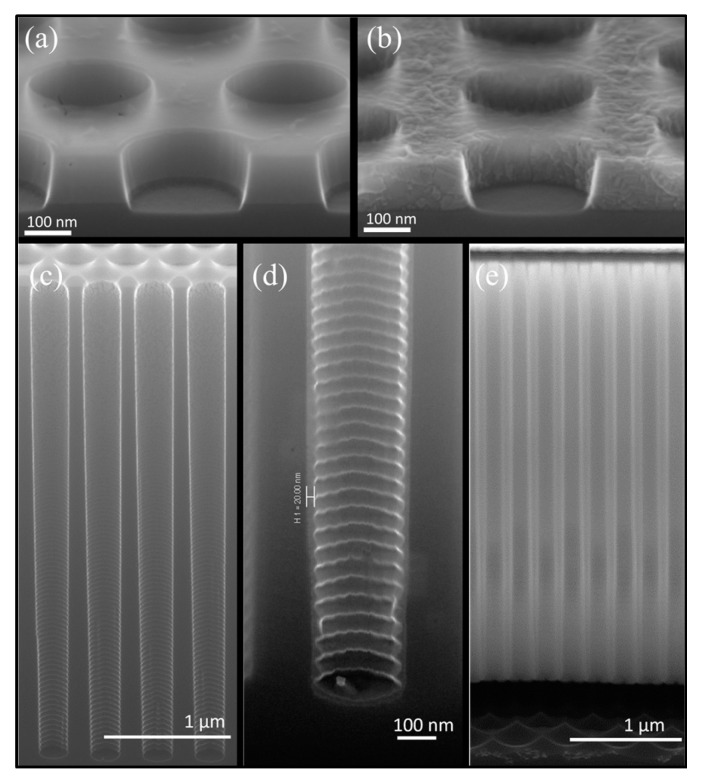
SEM images showing (**a**) patterned nano-pores in alumina; (**b**) patterned nano-pores in alumina after rapid thermal oxidation; (**c**) nano-pores of 200 nm opening diameter with 400 nm pitch, where $W_{top}=$ 280 nm is the maximum width at ca. the top of the nano-pore, and $d_{max}=$ 3.75 µm is the maximum depth of nano-pores, giving $AR=d_{max}/W_{mid}=$ 13.3; (**d**) a close-up view of the nano-pore bottom with a 20 nm uniform oxide layer grown on the sidewalls via rapid thermal oxidation; and (**e**) the formation of a 500 nm buried cavity underneath the membrane.

**Table 1 micromachines-16-00104-t001:** Optimized parameters for high-aspect-ratio nano-pore etching using the DREM sequence on a 150 mm Si wafer. The symbol “→” indicates the ramping of the parameter to prevent non-uniformities by ensuring consistent etch depth per cycle independent of the aspect ratio.

DREM Sequence 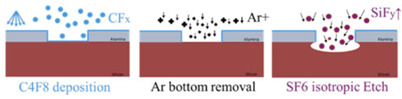		Optimized Process		
		123 Cycles		
		Deposit	Remove	Etch
**Duration (s)**		2 → 9	1.5 → 5.5	2.1 → 31.5
**Gas flows (sccm)**	**C4F8**	65 → 85	10	10
	**SF6**	15	20	50
	**Ar**	75	75	75
**13.56 MHz generators (W)**	**Coil power**	2000	2000	2000
	**Platen power**	1	65 → 85	1
**Chamber pressure (mTorr)**	**Pressure**	4.5	4.5	17

**Table 2 micromachines-16-00104-t002:** Parameter settings for through-silicon-via formation using the DREM sequence on a 150 mm Si wafer.

DREM Sequence 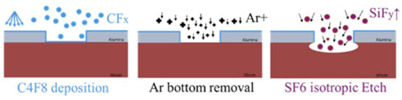		Optimized Process		
		123 Cycles		
		Deposit	Remove	Etch
**Duration (s)**		6	2	5
**Gas flows (sccm)**	**C4F8**	200	0	0
	**SF6**	0	350	550
**13.56 MHz generators (W)**	**Coil power**	2000	2000	2000
	**Platen power**	0	140	30
**Chamber pressure (mTorr)**	**Pressure**	25	25	150

## Data Availability

The original contributions presented in this study are included in the article. Further inquiries can be directed to the corresponding author.

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
