# Peer review of "Nano-Perforated Silicon Membrane with Monolithically Integrated Buried Cavity"

_micromachines, 2025, doi:10.3390/mi16010104_

Round 1
Reviewer 1 Report
Comments and Suggestions for Authors
Interesting scale-down of buried channel technology.
One major issue: after discussing DREM-version in length, the Outlook section introduced a seemingly better process for fabricating NPMs. It kind of undermines the results of the manuscript.
Technical comments:
page 4, line 120: I would start by saying that a hexagonal grid of 300 nm holes are patterned, and only later tell that chip size is 7 mm.
p4,L122: "selectively removed" has the connotation that they are removed by etching selectively, but they are removed in CAD-design phase.
P4,L129 and P6,L172: different alumina etch processes are used for NPM and TSV fabrication without any explanation.
Table 1: I would write Deposit, Removal, Etch out, and not rely on all people remembering DREM. Same applies to Table 2, too.
P7,L205 and Fig. 4: silicon nitride is a strange choice for DRIE mask since nitride and silicon are both etched by SF6. What kind of selectivities can be achieved?
Fig. 4d: if indeed FC step coverage is as depicted, wouldn't it be beneficial for R-step, since there is no FC to remove from bottom?!
Fig.4d profile is not universal: there are FC deposition processes that result in conformal coverage.
P9,L249: PECVD oxide is first said to be conformal, and then, on Line 259 and L266 it is said to be non-conformal.
P10,L267: PECVD process that is depositing nearly nothing on sidewalls is a very special case, and it should be described in detail.
P10,L274 discusses Si etch rate, but no mention is made of "up-etching": isotropic process proceeds also upwards to some extent. Some comments are made on P15,L451, but they could be better on P10.
Fig. 6g: almost nothing can be inferred from SEM. This is such an exotic additional feature of the proposed process that it deserves a schematic figure of its own.
P12 Thermal oxide: cleaning the wafer after DRIE before thermal oxidatation requires careful cleaning, which is not explained, but in Outlook the issue is touched and admitted: P15,L440.
P13,L361: PECVD oxide etch rate is 30-35 nm/min, and calculating from 465 s etch time, bottom oxide etch rate is only 2.5 nm/min. This huge discrepancy deserves some discussion.
P15,L442: this is a novel version of the LOCOS-process, using alumina instead of silicon nitride as a mask in thermal oxidation. It should certainly be discussed more, e.g. what does alumina crystallization (which happens at ca. 900oC) mean to the process (assuming that CORE-version of the process is part of the final paper).
Author Response
One major issue: after discussing DREM-version in length, the Outlook section introduced a seemingly better process for fabricating NPMs. It kind of undermines the results of the manuscript.
Response: We understand your concern that the simpler CORE process may take some steam out of the main part of the article. However, the CORE process proposed in the outlook section describes an alternative fabrication path. This alternative process cannot simply be categorized as a “better” process. As with any method, there are pros and cons. One of the challenges we faced with the CORE process, as described here, was the limitation of the rapid thermal oxidation step. This approach constrained us to achieve only a 20 nm oxide layer for sidewall passivation, which proved insufficient for thicker NPM structures, as previously mentioned. This limitation resulted in reduced reliability compared to the DREM approach, despite the CORE process offering other advantages. The main advantages are simplicity and avoiding fluorocarbon chemistry for sidewall passivation in etching nanopores. We added the outlook section in order not to publish incremental results, but rather give the honest full picture, and let the readers, who may want to make use of our results, choose by themselves which process to employ.
Technical comments:
Comments 1. page 4, line 120: I would start by saying that a hexagonal grid of 300 nm holes are patterned, and only later tell that chip size is 7 mm.
Response 1: We have revised the description to improve clarity and precision. “A hexagonal array of circular nano-pores, each with a diameter of 300 nm and a pitch of 700 nm, is patterned onto the resist using an exposure dose of 370 J/m² with zero focus offset. Following exposure, the resist is developed in AZ 726 MIF (MicroChemicals) for 60 s. This process results in a 7 mm wide perforated circular area featuring patterned nano-pores. Page 5, L127-130.”
Comments 2. p4, L122: "selectively removed" has the connotation that they are removed by etching selectively, but they are removed in CAD-design phase.
Response 2: We have revised the text to improve clarity and detail about the pore-free regions and their arrangement. “Pore-free regions of each 10 µm in diameter are intentionally designed created in this circular perforated area, and they are repeated in hexagonal arrangement with 50 µm pitch to serve as support structures for the NPM.” Page 5, L131-133.
Comments 3. P4,L129 and P6,L172 different alumina etch processes are used for NPM and TSV fabrication without any explanation.
Response 3: We agree with the comment. Therefore, we have updated the text for clarity. “The pattern transfer into the alumina layer uses a 30 sccm BCl3 gas flow at 1 mTorr, -20 °C, 30 W platen power, and 450 W coil power for 240 s. The low platen power is specifically optimized to minimize resist erosion (only 360 nm thick) and achieve a straight wall profile in the alumina layer. Page 5, L139-142.
“Subsequently, the developed pattern was transferred into the alumina layer using chlorine plasma as previously described.” Page 6, L183-184”
Comments 4. Table 1: I would write Deposit, Removal, Etch out, and not rely on all people remembering DREM. Same applies to Table 2, too.
Response 4: Thank you for pointing this out. We agree with your observation and have made the necessary changes in both Table 1 and Table 2.
Comments 5. P7,L205 and Fig. 4 Silicon nitride is a strange choice for a DRIE mask since nitride and silicon are both etched by SF6. What kind of selectivity can be achieved?
Response 5: We acknowledge that silicon nitride is used as a masking layer for RIE in [16] formerly as [17]. While both SiN and Si are etched by SF6, SiN etching requires ions to proceed and does not occur spontaneously. In our manuscript, the reference to SiN is based on the work of De Boer et al. (cited in reference 17) and is mentioned as one of several strategies reported in the literature for effective top surface protection during etching. However, we emphasize that this material is not used in the present study. “Several techniques might be helpful to get better top protection, and three are mentioned in [16], Figure 4(a-c). However, these proposals all rely on an LPCVD SiNx or other mask still present on top of the pores. In the revised manuscript the citation number is changed to [16].” Page 8, L218-220
Comments 6. Fig. 4d: if indeed FC step coverage is as depicted, wouldn't it be beneficial for R-step, since there is no FC to remove from bottom?!,
Response 6: We appreciate your observation regarding the implications of FC ste coverage for the R-step. To address this, we have carefully modified Figure 4d to more accurately depict the FC coverage including the PECVD deposition coverage. Page 8, L227.
Comments 7. Fig.4d profile is not universal: there are FC deposition processes that result in conformal coverage
Response 7: We understood that the Fig. 4d profile may not represent all FC deposition processes, as some indeed result in conformal coverage. To address this, we have modified the caption for Fig. 4d to clarify that the depicted profile represents a specific scenario and is not universal. “Three methods a-c from [16] to protect the coating in the trench and two alternative ways d-e used in this study for nano-pores. (a) Under etching plus thermal oxidation, (b) Isotropic pre-etching, (c) Short sacrificial layer etching, (d) Non-conformal fluorocarbon deposition in DRIE etch tool for nano-pores, (e) Thermal oxidation topped with non-conformal PECVD oxide in nano-pores.” Page 8, 228-231.
Comments 8. P9,L249: PECVD oxide is first said to be conformal, and then, on Line 259 and L266 it is said to be non-conformal
Response 8: L259: Upon review, we confirm that the text consistently describes PECVD oxide as non-conformal. On Line 259, the manuscript explicitly distinguishes PECVD oxide as a non-conformal technique, contrasting it with conformal methods.“To create SiO2 protection, many techniques are available that either form conformal (e.g. thermal, chemical, plasma, or LPCVD oxide) or non-conformal (e.g. spin on glass or PECVD oxide) films.” Page 9, L257-259.
Comments 9. P10,L267: PECVD process that is depositing nearly nothing on sidewalls is a very special case, and it should be described in detail
Response 9: This misconception of no deposition on sidewalls has already been addressed in our response to Comments 6 and 7. To ensure consistency, we have updated the text in the manuscript to clarify the context of the PECVD process. Moreover, this specific step has already been discussed in detail in the manuscript, with supporting explanations provided in Figure 8. “The double protection continues with PECVD oxide deposition, which is inherently non-conformal in nano-pores due to ion-enhanced reactions at high pressure (700 mTorr), resulting in frequent collisions that reduce the reactive species mobility and hinder the diffusion of reactive species into narrow spaces. As a result, more oxide is deposited near the pore opening, while the sidewalls and bottom of the pore have much less oxide, as shown in Figure 4(e).” Page 10, L274-279.
Comments 10. P10,L274 discusses Si etch rate, but no mention is made of "up-etching": isotropic process proceeds also upwards to some extent. Some comments are made on P15,L451, but they could be better on P10.
Response 10: Thank you for pointing this out. Indeed, regarding upward etching during cavity formation there is upward etching as well due to the isotropic nature of SF₆ etching.
However, this has already been addressed in the manuscript under the section "Fabrication Challenges Affecting NPM Integrity," where upward undercutting is discussed. To improve clarity, Figure 2 has been updated to explicitly illustrate the upward undercutting, similar to what is shown in Figure 7. We believe this enhancement provides a better understanding of the process.
Comments 11. Fig. 6g: almost nothing can be inferred from SEM. This is such an exotic additional feature of the proposed process that it deserves a schematic figure of its own.
Response 11: Thank you for your valuable suggestion. To address this, we have added a detailed schematic in Figure 7 including a detailed explanation. Page 11, L303-321.
Comments 12: P12 Thermal oxide: cleaning the wafer after DRIE before thermal oxidation requires careful cleaning, which is not explained, but in Outlook the issue is touched and admitted: P15,L440.
Response 12: We confirm that the cleaning procedure is mentioned in the Materials and Methods section (L142), where we describe the use of RCA cleaning for the wafer prior to thermal oxidation. Page 6, L154. “This is followed by another ashing process as previously described to remove residues caused by the Si etch. The remaining alumina is removed by etching in 10% hydroflu-oric acid (HF) for 60 s followed by RCA cleaning.“
Comments 13: P13,L361: PECVD oxide etch rate is 30-35 nm/min, and calculating from 465 s etch time, bottom oxide etch rate is only 2.5 nm/min. This huge discrepancy deserves some discussion.
Response 14: The discrepancy in etch rates—30-35 nm/min for PECVD oxide versus only 2.5 nm/min for the thermal oxide at the nanopore bottom—can be attributed to geometric and transport limitations (RIE lag) within the nanopores. Moreover, this specific step has already been discussed in detail in the manuscript, with supporting explanations provided in Figure 8. To enhance clarity, we have refined the text. “The difference in the etch rate of surface and bottom oxide is due to RIE lag in the HAR structure [22]. Additionally, reduced nano-pore opening caused by the PECVD oxide layer (Figure 8(a)) resulted in a lower initial etch rate for the bottom oxide." Page 14, L386-389.
Comments 14: P15,L442: this is a novel version of the LOCOS-process, using alumina instead of silicon nitride as a mask in thermal oxidation. It should certainly be discussed more, e.g. what does alumina crystallization (which happens at ca. 900oC) mean to the process (assuming that CORE-version of the process is part of the final paper).
Response 15: Indeed, this process is a local oxidation of silicon (LOCOS), and resembles the classical LOCOS process aimed at the electrical separation of complementary transistor pairs in CMOS technology, although in this case, the geometry is vastly different. We acknowledge that alumina crystallinity is expected to emerge above approximately 800-900°C. However, we do not anticipate that the rapid thermal oxidation (RTO) step will compromise the integrity of the top alumina layer. To address your concern, we have added a more detailed discussion of this aspect in the revised manuscript, including an SEM image of the alumina layer after RTO to visually support our findings.“Figure 11(a) and (b) show the nanopore-patterned alumina before and after RTO. After RTO, the amorphous alumina is transformed into crystalline alumina, and a transition usually happens above 800-900°C. This change could increase the porosity of the alumina mask layer due to nanocrystal formation. However, the effect is more significant in thin films (2-7 nm) [23] than in thicker layers like the 100 nm alumina used here. Moreover, this change did not have an observed effect on the etching of Si in our experiments.” Page 17, 494-500.

Reviewer 2 Report
Comments and Suggestions for Authors
PFA

Author Response
Comments 1. The limitation of Boer. et al’s work should be written clearly on page 2.
Response 1: We agree that clearly stating the limitations of Boer et al.’s work is essential for providing proper context. We have revised the text on page 2 to explicitly outline the limitations of their study and to describe how this study addresses those issues. “However, this method cannot be directly applied in our cleanroom due to the potential contamination of oxide or nitride furnaces by wafers previously processed with metal or metal oxides. As a result, the hard mask materials must be removed after feature (trench or pore) etching (explained later in more detail). Due to these constraints, an alternative approach is necessary to effectively protect the top surfaces and edges while selectively removing the bottom protection layer to form buried channels or cavities. By introducing modifications to pore arrangements and adapting fabrication steps, our method enables the formation of buried cavities with integrated support structures using nano-pores. Additionally, the proposed approach offers flexibility for adaptation in other facilities with similar contamination sensitivity.” Page 3, L84-93.
Comments 2. Emphasis should be provided on the complexity of the etching technique being used which will turn out to be a big limiting factor during the scale up of the technology. While termed "wafer-scale," the practicality of scaling this method for industrial production remains unclear and could benefit from further exploration.
Response 2: We do not agree with the reviewer that the complexity of the etching technique will turn out to be a big limiting factor for scaling up the process. “Wafer-scale" refers to the ability to implement the fabrication process reproducibly on entire wafers. We demonstrated this by fabrication on entire 150 mm wafers in our tests; We utilized standard fabrication technologies to ensure that this approach can be adapted and scaled universally with tailored adjustments to meet specific requirements.
Comments 3. Although a wide range of applications is proposed, the article could provide more in-depth experimental validation for specific industrial or biomedical scenarios.
Response 3: To address your comment, we have expanded the conclusion section to briefly elaborate on the potential applications of this technology. However, providing a detailed explanation of the application lies beyond the scope of this paper, which is primarily focused on addressing the challenges and advancements in nanofabrication. “For instance, as an interface for mass spectrometry, NPM serves as a capillary stop type of liquid/gas interface facilitating the separation of volatile gas molecules dissolved in the bulk liquid, which can then be analyzed using mass spectrometry.” Page 16, L464-467.
Comments 4. The robustness of the membranes under prolonged operational conditions, such as high-pressure or corrosive environments, warrants additional investigation.
Response 4: We understand your concern however, as stated above, this paper focuses on the fabrication issues, and we have thus not included any functional testing, as this would fall out of scope for this paper and would require a lot more time for extended experiments.

Reviewer 3 Report
Comments and Suggestions for Authors
Overall a very clearly presented paper on the process flow and optimizations/challenges of making a monolithic NPM on silicon substrates. Here are some comments to improve the quality of the paper further:
1. Some more quantitative details can be provided on the mechanical properties of the NPM, possibly simulations or even better, some experimental measurements of strength/integrity under various conditions.
2. Experimental demonstration of an application of the fabricated monolithic NPM would add more value and legitimacy to the study.
Author Response
Comments 1. Some more quantitative details can be provided on the mechanical properties of the NPM, possibly simulations or even better, some experimental measurements of strength/integrity under various conditions.
Response 1: Thank you for your suggestion. This paper primarily focuses on the fabrication challenges of the nanopore membrane (NPM), and functional testing, including quantitative mechanical properties, is not included as it would fall outside the scope of this study. Conducting such experiments would require additional time and extensive experimental setups. However, to address your comment and provide a foundational discussion on the mechanical properties of the NPM, we have added a new section, 3.5 Mechanical Strength of NPM, on Page 16, L 428-446
“3.5 Mechanical strength of NPM
The mechanical strength of the NPM is a critical characteristic and can be evaluated using various methods. For fluidic applications such as separation and filtration, the mechanical strength can be effectively represented by the transmembrane pressure ( ). The maximum transmembrane pressure before membrane with a radius, and membrane thickness , the fracture can be estimated using the following equation [10],
Here is the yield stress and is Young’s modulus of Si. is the non-perforated fraction of NPM (~0.5). However, while Young’s modulus of Si is well known (130 GPa), the yield stress may vary orders of magnitude, and in addition depends on processing conditions. Hence, by choosing a very conservative (low) estimate for the yield strength of 165 MPa, a lower bound for 2 bar is obtained, where also the radius of 45 µm is inserted for instead of the 3.5 mm (actual membrane radius). 45 µm is the estimated radius of the largest unsupported region, where the support structures are removed for TSV etching. At this unsupported region, the membrane is more prone to fracture.”
Comments 2. Experimental demonstration of an application of the fabricated monolithic NPM would add more value and legitimacy to the study
Response 2: As this study focuses solely on the fabrication process, demonstrating an application would require additional steps and resources, which are currently beyond the scope of this work. However, we have briefly elaborated on the potential applications of the fabricated NPM in the conclusion section, highlighting its relevance to mass spectrometry. This addition aims to provide a broader perspective on the significance of this work and its potential for future exploration. “For instance, as an interface for mass spectrometry, NPM serves as a capillary stop type of liquid/gas interface facilitating the separation of volatile gas molecules dissolved in the bulk liquid, which can then be analyzed using mass spectrometry.” Page 16, L464-467.

Round 2
Reviewer 1 Report
Comments and Suggestions for Authors
One small detail: Cr hard mask limits further wafer processing, but Al2O3 hard mask does not seem to be an obstacle to thermal oxidation later in the process (in both cases metal-containing masks have been removed before further processing).
Author Response
Comment: "One small detail: Cr hard mask limits further wafer processing, but Al2O3 hard mask does not seem to be an obstacle to thermal oxidation later in the process (in both cases metal-containing masks have been removed before further processing)."
Answer: Yes, we remove the alumina mask after the etch step in our process. The alumina mask is removed because the next step involves RCA and thermal oxidation, and our cleanroom protocol does not permit the presence of metals or metal oxides in the thermal furnace. Therefore, alumina is eliminated to comply with these restrictions. In contrast, the reference article retained the Cr mask (any metal or metal oxide mask) during thermal oxidation (somehow). As a result, the BCT approach described in their study is not feasible in our cleanroom or other cleanrooms with similar constraints. To address this, we developed an alternative approach for fabricating NPMs, which works effectively even after the removal of the mask (whether it is a metal or metal oxide). We have already addressed this Pg 2 L86-90.